# Repurposing Anti-diabetic Drugs to Cripple Quorum Sensing in *Pseudomonas aeruginosa*

**DOI:** 10.3390/microorganisms8091285

**Published:** 2020-08-22

**Authors:** Wael A. H. Hegazy, Maan T. Khayat, Tarek S. Ibrahim, Majed S. Nassar, Muhammed A. Bakhrebah, Wesam H. Abdulaal, Nabil A. Alhakamy, Mahmoud M. Bendary

**Affiliations:** 1Department of Microbiology and Immunology, Faculty of Pharmacy, Zagazig University, Zagazig 44519, Egypt; 2Department of Pharmaceutical Chemistry, Faculty of Pharmacy, King Abdulaziz University, Jeddah 21589, Saudi Arabia; mkhayat@kau.edu.sa (M.T.K.); tmabrahem@kau.edu.sa (T.S.I.); 3Center of Excellence for Drug Research & Pharmaceutical Industries, King Abdulaziz University, Jeddah 21589, Saudi Arabia; nalhakamy@kau.edu.sa; 4Department of Pharmaceutical Organic Chemistry, Faculty of Pharmacy, Zagazig University, Zagazig 44519, Egypt; 5Life Science and Environment Research Institute, King Abdulaziz City for Science and Technology (KACST), P.O. Box 6086, Riyadh 11442, Saudi Arabia; mnassar@kacst.edu.sa (M.S.N.); Mbakhrbh@kacst.edu.sa (M.A.B.); 6Department of Biochemistry, Cancer Metabolism and Epigenetic Unit, Faculty of Science, King Abdulaziz University, Jeddah 21589, Saudi Arabia; whabdulaal@kau.edu.sa; 7Department of Pharmaceutics, Faculty of Pharmacy, King Abdulaziz University, Jeddah 21589, Saudi Arabia; 8Department of Microbiology and Immunology, Faculty of Pharmacy, Port Said University, Port Said 42526, Egypt; m.pendary@pharm.psu.edu.eg

**Keywords:** *Pseudomonas aeruginosa*, quorum sensing, virulence, sitagliptin, metformin, vildagliptin

## Abstract

*Pseudomonas aeruginosa* is a significant human pathogen, it possesses almost all of the known antimicrobial resistance mechanisms. Quorum sensing (QS) is an intercellular communication system that orchestrates bacterial virulence and its targeting is an effective approach to diminish its pathogenesis. Repurposing of drugs is an advantageous strategy, in this study we aimed to repurpose the anti-diabetic drugs sitagliptin, metformin and vildagliptin as anti-QS in *P. aeruginosa*. The effects of sub-inhibitory concentrations of the tested drugs on the expression of QS-encoding genes and QS-regulated virulence factors were assessed. The protective activity of tested drugs on *P. aeruginosa* pathogenesis was evaluated in vivo on mice. In silico analysis was performed to evaluate the interference capabilities of the tested drugs on QS-receptors. Although the three drugs reduced the expression of QS-encoding genes, only sitagliptin inhibited the *P. aeruginosa* virulence in vitro and protected mice from it. In contrast, metformin showed significant in vitro anti-QS activities but failed to protect mice from *P. aeruginosa*. Vildagliptin did not show any in vitro or in vivo efficacy. Sitagliptin is a promising anti-QS agent because of its chemical nature that hindered QS-receptors. Moreover, it gives an insight to consider their similar chemical structures as anti-QS agents or even design new chemically similar anti-QS pharmacophores.

## 1. Introduction

Antimicrobial resistance is an increasing problem, particularly in the dwindling discovery of new safe and eligible antibiotics [1]. Curtailing of bacterial virulence is an interesting alternative strategy for antibiotic therapy [2]. Quorum sensing (QS) is the process in which bacterial populations are governed by cell–cell communication through signaling molecules produced by bacterial cells called autoinducers that are detected by receptors on other cells. The QS signaling system controls diverse physiological functions in both Gram-positive and Gram-negative bacteria [3]. Dozens of studies describe how the clinically-relevant bacteria utilize QS to control the collective production of virulence factors and the efforts to inhibit QS in these pathogens with the aim of designing new efficient antimicrobial therapeutics [3,4,5]. In this direction, we and several groups studied the ability of several compounds and chemical moieties to serve as anti-biofilm and anti-QS agents, giving a great consideration to these chemical compounds as their safeties have been already approved [6,7].

Discovering new applications for approved drugs other than their renowned uses is a well-known strategy called the repurposing of drugs. This strategy of drug repurposing is progressively introduced as an elegant proposition, as it exhibits several merits. The repurposed drugs’ safeties have advantageously been pre-approved and the formulation studies have already been accomplished. Moreover, this strategy reduces the costs and the time required for new drug development [8]. The importance of the drug repurposing strategy may be more important when these drugs are as routinely used as those prescribed for chronic diseases. Anti-diabetic drugs are daily used around the world, metformin is one of these drugs which is widely prescribed and even evaluated as the most common prescribed individual drug among oral hypoglycemic agents [9,10]. Metformin, with its biguanide chemical structure, showed activity against *P. aeruginosa* [11]. Furthermore, metformin potentiated the antibacterial activities of gold nanoparticles and augmented the biofilm eradication [12]. Significantly, we showed in a previous work that a pyrazine derivative with triazole moiety anti-diabetic drug sitagliptin can be repurposed as a virulence mitigating agent in *Serratia marcescens* [6]. Vildagliptin is another oral anti-diabetic containing pyrrolidine chemical moiety that was proven to have antifungal and antibacterial activity against *P. aeruginosa*, showing anti-QS activity [13].

*Pseudomonas aeruginosa* is an ubiquitous opportunistic Gram-negative bacterium capable of infecting virtually all tissues, causing both acute and chronic infections in humans [3]. *P. aeruginosa* is considered one of the most frequent nosocomial pathogens that can cause surgical wounds, burn wounds, lung, eye, bloodstream and urinary tract infections [14,15]. *P. aeruginosa* uses QS in production of an arsenal of virulence factors that augment its pathogenesis [3]. Because of the high resistance of *P. aeruginosa* and its serious infections that may be fatal especially for immunocompromised patients, QS in *P. aeruginosa* has been extensively studied [3,15]. The low efficacy of anti-pseudomonal therapies may be owed to its exceptional ability to evolve heritable resistance to various antibiotics, and its ability to establish surface-associated biofilm infections that defeat both host immune defense and antibiotics [16]. QS regulates the virulence of *P. aeruginosa*, and as a consequence targeting of QS has been proposed as an attractive alternative strategy is that targeting the pathogenesis of *P. aeruginosa* [3,17,18,19]. Another merit of this strategy is that targeting QS avoids direct effects on bacterial growth that may both enhance the development of bacterial resistance and the destruction of the normal flora in gut [20,21]. *P. aeruginosa* causes severe infections in diabetic “immunocompromised” patients and constitutes a real obstacle to be resolved, especially in wound infections and diabetic foot infections [22,23]. Bearing in mind the approved anti-virulence efficiency of anti-diabetic sitagliptin against *Serratia marcescens* [6], we aimed to investigate more anti-diabetic drugs against *P. aeruginosa*. In the current study we aimed to investigate the possibility of repurposing the chemically diverse anti-diabetic drugs sitagliptin, vildagliptin and metformin as inhibitors of *P. aeruginosa* QS in-vitro and in-vivo.

## 2. Materials and Methods

### 2.1. Bacterial Strains and Chemicals

Experiments were conducted using the model strain *Pseudomonas aeruginosa* PAO1 (ATCC BAA47B1) and *Chromobacterium violaceum* CV026 ATCC 31,532 that was used in the violacein inhibition assay. Tryptone soy broth (TSB), Mueller Hinton (MH) broth and agar and Luria–Bertani (LB) broth and agar were the products of Oxoid (Hampshire, UK). Vildagliptin (CAS 274901-16-5), sitagliptin (CAS 654671-77-9) and metformin (CAS 1115-70-4) were purchased from Sigma-Aldrich (St. Louis, MO, USA). Other used chemicals were of pharmaceutical grade. The dissolved drugs in dimethyl sulfoxide (DMSO) were usually prepared prior to each experiment and if needed they were kept at room temperature in dark place.

### 2.2. Determination of Minimum Inhibitory Concentration (MIC) of Tested Drugs and Their Sub-MIC Effect on Bacterial Growth

The MICs of the tested drugs were determined using the agar dilution method according to the Clinical Laboratory and Standards Institute Guidelines (CLSI, 2015). To avoid any effect on the bacterial growth, sub-inhibitory concentrations (1/10 MIC) of drugs were used to evaluate their inhibitory effects on *P. aeruginosa* virulence and QS. For standardizing the results in all the performed experiments, the bacterial cultures treated or not treated with 1/10 MIC of the tested drugs were adjusted to a fixed growth density of approximately 1 × 10^8^ CFU/mL (OD600 of 0.4). It is worthy to mention that the final concentration of DMSO solvent in the used media in all the following experiments was 2% which has no significant effect on *P. aeruginosa* virulence.

The effects of sub-inhibitory concentrations of the tested drugs on the growth of *P. aeruginosa* were identified according to Nalca et al. [24]. Adjusted *P. aeruginosa* cultures to 0.5 McFarland Standard were used to inoculate LB broth with 1/10 MIC of the tested drugs or control LB broth without the tested drugs and the optical densities were adjusted approximately to an OD600 of 0.4. After overnight incubation, the optical densities of both cultures with or without sub-MIC of drugs were measured at 600 nm using a Biotek Spectrometer (Biotek, Winooski, VT, USA). The experiment was carried out three times and data are presented as the mean ± standard error.

### 2.3. Violacein Inhibition Assay

The inhibitory effect of anti-diabetic drugs on the production of the quorum sensing violacein pigment was estimated according to Choo et al. [25]. Overnight cultures of *Chromobacterium violaceum* CV026 were prepared and adjusted approximately to an OD600 of 1. Aliquots of 100 μL of LB broth containing *N*-hexanoyl homoserine lactone with and without 1/10 MIC of anti-diabetic drugs were added to the wells of a 96-well microtiter plate that were provided with 100 μL aliquots of the bacterial suspensions. The plates were incubated at 28 °C for 16 h and were completely dried at 60 °C. Violacein pigment was extracted from wells by 100 μL of dimethyl sulfoxide (DMSO) and incubated at 30 °C with shaking. A negative control using DMSO alone was prepared. Violacein was quantified by measuring the absorbance at 590 nm. The experiment was done in triplicate and the inhibitory effect on violacein production of anti-diabetic drugs in sub-MIC treated cultures were expressed as mean ± standard error of percentages compared to untreated control cultures. The violacein production was compared with the untreated controls and the percentages of inhibition were calculated according to the formula:

[(OD590 control − OD590 in presence of tested drug in sub-MIC)/OD590 control] × 100

### 2.4. Biofilm Inhibition Assay

The ability of tested anti-diabetic drugs to inhibit biofilm formation was analyzed using the microtitre dish biofilm formation assay [6,26]. Aliquots of 10 μL of the *P. aeruginosa* suspension were adjusted approximately to OD600 of 0.4 and added to 1 mL amounts of fresh TSB with and without 1/10 MIC of the tested drugs. TSB Aliquots (100 μL) with or without sub-MIC of tested drugs were introduced into the microtiter plates wells and incubated overnight at 37 °C. The unadhered planktonic cells were sucked away and the wells were washed with distilled water and air dried. The adhered cells were fixed with methanol for 25 min and stained with 1% crystal violet for 30min. The wells were washed triplicate and rinsed by glacial acetic acid (33%). The experiment was repeated three times and the absorbance was measured at 590 nm using. The absorbance of anti-diabetic drugs treated bacteria were presented as mean ± standard error of percentage change from untreated bacterial controls. The biofilm inhibition percentages were calculated according to the formula:

[(OD590 control − OD590 in presence of tested drug in sub-MIC)/OD590 control] × 100

### 2.5. Motilities Inhibition Assay

The abilities of anti-diabetic drugs to interfere in the *P. aeruginosa* swimming, swarming and twitching motilities were investigated [6,27,28]. For swimming or swarming assay, 0.3% or 0.5% LB agar plates with or without the tested drugs in sub-MIC were point inoculated with 5 µL of the bacterial suspension and the plates were incubated at 37 °C for 24 h. For the twitching assay, 1% LB agar plates with or without the tested drugs in sub-MIC were centrally stabbed with 5 μL of the *P. aeruginosa* suspensions and incubated for 48 h at 37 °C. Then, the agar was removed, the plates were left to dry and the twitching zones were stained with crystal violet (1%). After rinsing the dye with water, the twitching zones were measured. Plates with DMSO were prepared and inoculated with *P. aeruginosa* in the same way as negative control. The experiment has been repeated in triplicate and the zones of swimming, swarming or twitching of *P. aeruginosa* treated with anti-diabetic drugs compared to untreated *P. aeruginosa* were measured in mm and showed as mean ± standard error of percentage change from untreated bacterial controls. The percentages of swimming, swarming or twitching inhibition were estimated using the formula:

[(Motility zone diameter of control − Motility zone diameter in presence of anti-diabetic drugs in sub-MIC)/Motility zone diameter of control] × 100

### 2.6. Protease Inhibition Assay

To evaluate the protease inhibitory activities of anti-diabetic drugs, the skim milk agar method was used [6,29]. *P. aeruginosa* treated with anti-diabetic drugs (sub-MIC) or untreated cultures were adjusted to an OD600 of 0.4, centrifuged at 21,000× *g* for 20 min. Supernatants (100 μL) were added to the wells made in 5% skim milk agar plates. After overnight incubation, the clear zones diameters surrounding the bacterial growth were measured in mm. The experiment was performed in triplicate and the obtained clear zones due to protease produced by anti-diabetic treated bacterial cultures were presented as mean ± standard error of percentage change from those obtained by untreated bacterial controls. The percentages of protease inhibition were calculated according to the formula:

[(Clear zone diameter of control − Clear zone diameter in presence of anti-diabetic drugs in sub-MIC)/Clear zone diameter of control] × 100

### 2.7. Hemolysin Inhibition Assay

Hemolysin activities of anti-diabetic drugs in sub-MIC treated and untreated *P. aeruginosa* were assayed by the method of Rossignol et al. [30]. *P. aeruginosa* treated with anti-diabetic drugs (sub-MIC) or untreated cultures were adjusted to an OD600 of 0.4, centrifuged at 21,000× *g* for 20 min. Supernatants (500 µL) were added to fresh erythrocytes suspension (2%) in 0.8 mL saline, incubated for 2 h at 37 °C and centrifuged at 11,000× *g* for 10 min at 4 °C. A positive control (complete hemolysis) was prepared by adding 0.1% Sodium Dodecyl Sulfate to an erythrocyte suspension and a negative control of unhemolyzed erythrocytes was prepared by incubation of erythrocytes in LB broth under the same circumstances. The hemoglobin release was assessed by determining absorbance at 540 nm and the experiment was performed in triplicate. The hemolysis which occurred in anti-diabetic drugs in sub-MIC treated cultures was presented as mean ± standard error of percentage change from the hemolysis of untreated control cultures. The released hemoglobin was compared with the positive and negative controls and the percentage of hemolysis inhibition was calculated using the formula:

[(Anti-diabetic treated or untreated bacterial cultures − Negative control)/(Positive control − Negative control)] × 100

### 2.8. Elastase Inhibition Assay

The elastolytic inhibitory activities of anti-diabetic drugs, were performed according to Ohman et al. [31]. Elastin Congo Red (ECR) reagent was prepared, 10 mg of ECR in 0.5 mL buffer [0.1 mol/L Tris pH 7.2 and 10 mol/L CaCl_2_]. *P. aeruginosa* treated with anti-diabetic drugs (sub-MIC) or untreated overnight cultures were adjusted to an OD600 of 0.4, centrifuged at 21,000× *g* for 20 min. Supernatants (500 µL) were mixed with the ECR reagent, incubated with shaking for 6 h at 37 °C and centrifuged to remove the insoluble ECR. The absorbance of the Congo red dye was measured at 495 nm. The experiment was performed in triplicate and the elastolytic activity of anti-diabetic drugs in sub-MIC treated cultures was expressed as mean ± standard error of percentages compared to the elastase produced by untreated control cultures. The elastase activities were compared with the controls and the percentages of elastase production were calculated according to the formula:

[(OD495 control−OD495 in presence of tested drug in sub-MIC)/OD495 control] × 100

### 2.9. Pyocyanin Inhibition Assay

The assessment of the inhibitory effect of anti-diabetic drugs on formation the of the bluish-green pyocyanin pigment by *P. aeruginosa* was performed according to Das and Manefield [32]. *P. aeruginosa* overnight cultures were adjusted to an OD600 of 0.4 and 10 μL aliquots of the bacterial suspensions were added to 1 mL LB broth with or without anti-diabetic drugs in sub-MIC. The tubes were incubated at 37 °C for 48 h, centrifuged at 11,000× *g* for 10 min and the produced pyocyanin was measured at 691 nm. The experiment was performed in triplicate and the inhibitory effect on pyocyanin production of anti-diabetic drugs in sub-MIC treated cultures was expressed as mean ± standard error of percentages compared to untreated control cultures. The pyocyanin production was compared with the controls and the percentages of inhibition were calculated according to the formula:

[(OD691 control − OD691 in presence of tested drug in sub-MIC)/OD691 control] × 100

### 2.10. Inhibition Assay of Oxidative Stress Resistance

The disk assay method of Hassett et al. was used to evaluate the ability of the tested anti-daibetic drugs in sub-MIC to inhibit resistance of *P. aeruginosa* to oxidative stress [33]. Overnight cultures of *P. aeruginosa* in LB broth were adjusted to an OD600 of 0.4 and 0.1 mL aliquots were uniformly spread on the surface of LB agar plates containing 1/10 MIC of tested anti-diabetic drugs. Hydrogen peroxide (1.5%) 10 μL was added to sterile paper disks (9 mm) present on the surface of LB agar plates that were incubated at 37 °C for 24 h. Control plates without anti-diabetic drugs were prepared in the same manner. The experiment was performed in triplicate and the inhibition zones of *P. aeruginosa* treated with anti-diabetic drugs compared to untreated *P. aeruginosa* were measured in mm and expressed as mean ± standard error.

### 2.11. Quantitative RT-PCR of QS and Virulence Genes

#### 2.11.1. RNA Extraction

Anti-diabetic treated and untreated *P. aeruginosa* cultures were obtained by centrifugation (6000× *g* for 15 min at 4 °C). Collected bacterial pellets were re-suspended in 100 μL of Tris-EDTA buffer provided with lysozyme and kept for 5 min at 25 °C prior to lysis by RNA lysis buffer. An RNAeasy Mini Kit (Qiagen, Germany) was used to isolate and purify total RNA according to the manufacturer’s instructions. Any residual chromosomal DNA was removed by DNase. Concentrations of RNA were measured by a NanoDrop ND-1000 spectrophotometer at 260 nm and 280 nm and kept at −80 °C until use.

#### 2.11.2. Real Time PCR (qRT-PCR)

The influences of sub-MIC of anti-diabetic drugs on expression levels of genes that encode the autoinducers (AI) of the three QS systems *lasI*, *rhlI* and *pqsA* and their receptors *lasR*, *rhlR* and *pqsR* in *P. aeruginosa* [3,34,35] were characterized using the RT-qPCR. Total RNA (10 ng) from each sample, untreated *P. aeruginosa* and anti-diabetic drugs in sub-MIC treated *P. aeruginosa* were used for cDNA synthesis by reverse transcription using a high capacity cDNA Reverse Transcriptase kit (Applied Biosystem, Foster City, CA, USA). The cDNA was amplified by the Syber Green I PCR Master Kit (Fermentas, Waltham, MA, USA) in a 48-well plate using the Step One instrument (Applied Biosystem, Foster City, CA, USA) as follows: enzyme activation at 95 °C for 10 min followed by 40 cycles of 15 sec at 95 °C, 20 sec at 55–65 °C and 30 sec at 72 °C for the amplification step. Changes in the expression of each target gene were normalized relative to the mean critical threshold (CT) values of *rpoD* in *P. aeruginosa* as housekeeping genes by the 2^−ΔΔCt^ method [34]. Specific forward and reverse primers (1 μM) for each gene were used, the sequences for used primer are shown in Table 1. The experiment was performed in triplicate and the gene expressions of anti-diabetic drugs which treated *P. aeruginosa* were expressed as mean ± standard error of fold change from untreated *P. aeruginosa* untreated controls.

### 2.12. Mice Survival Assay

Ethical approval: This article does not include any studies with human participants. The institutional Review Board (ethical committee) at the Faculty of Pharmacy, Zagazig University approved animal experiments in this study. The procedures were performed in compliance with the ARRIVE guidelines, in agreement with the U.K. Animals (Scientific Procedures) Act, 1986 and related guidelines (ECAHZU, 23 August 2015).

The conferred protection by tested anti-diabetic drugs against *P. aeruginosa* pathogenesis was investigated by the mice survival in vivo model [36]. The cell densities of *P. aeruginosa* overnight cultures in LB broth with or without sub-MIC of anti-diabetic drugs were adjusted to approximately 1 × 10^8^ CFU/mL in phosphate buffer saline (PBS) and in LB broth with DMSO in the same concentrations that were used as solvent for anti-diabetic drugs. Seven groups of 3-week old healthy female albino mice (*Mus musculus*) with similar weights were used, each containing 5 mice. For the first group, mice were intraperitoneally injected with 100 μL of sitagliptin-treated *P. aeruginosa* in sterile PBS, the second group was injected with 100 μL of metformin-treated *P. aeruginosa*, the third group was injected with vildagliptin-treated *P. aeruginosa*, the fourth group was injected with 100 μL of DMSO-treated *P. aeruginosa* and the fifth group was injected with 100 μL of untreated *P. aeruginosa*. Moreover, two negative control groups were added, one group was injected with 100 μL of sterile PBS and the other group was left uninoculated. The survival of mice was reported every day over five successive days. The experimental mice were held in plastic cages with wood shave bedding, survived in normal feeding and conditions such as aeration, humidity (70% ± 5%), temperature (24 °C ± 2 °C) and a 12 h light/dark cycle.

### 2.13. Molecular Modeling Study of Anti-Diabetic Drugs Binding to QS Receptors

To explore the molecular interaction of anti-diabetic drugs with the QS receptors, a molecular docking study was conducted using the Glide docking engine within the Schrodinger molecular modeling suite [37]. The crystal structures of the QS receptors LasR, QscR and PqsR (PDB: 1RO5, 6CC0, 6MVN) were used as a receptor for ligand docking.

## 3. Results

### 3.1. Determination of MIC Anti-Diabetic Drugs and Their Effect on Bacterial Growth

Sitagliptin, metformin and vildagliptin inhibited the *P. aeruginosa* PAO1 growth at 16 mg/mL, 100 mg/mL and 20 mg/mL, respectively. The concentrations equivalent to 1/10 MIC were used to test the anti-virulence and anti-QS activities of sitagliptin, metformin and vildagliptin with 1.6 mg/mL, 10 mg/mL and 2 mg/mL. To exclude the effect of tested anti-diabetics on the bacterial growth, the optical densities of the bacterial suspension of overnight cultures were measured at 600 nm in the presence or absence of tested anti-diabetics in sub-MIC (1/10). The experiment was conducted three times and data are expressed as the mean ± standard error. One-way ANOVA (Graphpad Prism 8 software, San Diego, CA, USA) was employed to test the statistical significance. The results were considered significant when *p* value < 0.05. There were no observed significant differences between the turbidity of the bacterial suspensions with or without anti-diabetic drugs in their sub-MIC, indicating the absence of their effects in sub-MIC on bacterial growth (Figure 1).

### 3.2. Inhibition Violacein Pigment Production

*C. violaceum* CV026 was used as a biosensor for quantitative evaluation of the QS activity, as it produces the purple violacein pigment only in the presence of acylhomoserine lactones in the growth media which can be quantified by spectrophotometry. The experiment was conducted in triplicate and the one-way ANOVA test (Graphpad Prism 8 software) was used to compare between absorbance values of sitagliptin, metformin and vildgliptin in sub-MIC treated and untreated *C. violaceum*. The results were considered significant when *p* values < 0.05. The data obtained were expressed as mean ± standard error of percentage change from untreated *C. violaceum* CV026 controls (Figure 2A). Sitagliptin and metformin showed a significant ability to inhibit violacein production (*p* < 0.0001) with inhibition percentages of 67% and 57%, respectively. Moreover, sitagliptin inhibited violacein production significantly when compared to metformin (*p* = 0.0245). On the other hand, vildagliptin was not able to inhibit the production of violacein significantly.

### 3.3. Inhibitory Effect of Tested Anti-Diabetic Drugs on the P. aeruginosa Virulence Factors

#### 3.3.1. Inhibition of Biofilm Formation

The biofilm production was quantified to demonstrate the capability of the tested anti-diabetic drugs in sub-MIC to interfere with biofilm formation. The experiment was performed in triplicate and the significance of mean difference between sitagliptin, metformin and vildagliptin in sub-MIC treated and untreated *P. aeruginosa* was attested using the one-way ANOVA test (Graphpad Prism 8 software). The results were considered statistically significant when *p* values < 0.05. The obtained results were expressed as mean ± standard error of percentage change from untreated *P. aeruginosa* control (Figure 2B). While vildagliptin did not inhibit the biofilm formation significantly, both sitagliptin and metformin inhibited significantly the formation of biofilm with inhibition percentages of 60% and 67%, respectively.

#### 3.3.2. Inhibition of *P. aeruginosa* Motilities

The diameters of *P. aeruginosa* swimming, swarming and twitching on LB agar plates with or without anti-diabetic drugs in sub-MIC were measured. The experiments were repeated three times, and the one-way ANOVA test (Graphpad Prism 8 software) was used to compare between sitagliptin, metformin and vildagliptin in sub-MIC treated and untreated *P. aeruginosa*, and the results were considered statistically significant when *p* values < 0.05. Significantly, sitagliptin and metformin decreased *P. aeruginosa* swimming, swarming and twitching with inhibition percentages (56%, 67% and 78%) and (44%, 55% and 69%), respectively. Bear in mind that sitagliptin reduced the bacterial motilities more significantly than metformin. The data were expressed as mean ± standard error of percentage change from untreated *P. aeruginosa* controls (Figure 2C–E). Importantly, vildagliptin did not inhibit any of the *P. aeruginosa* motilities.

#### 3.3.3. Inhibition of *P. aeruginosa* Virulence Enzymes

Enzymes as protease, hemolysin and elastase enzymes are regarded as virulence factors contributing to the pathogenicity of *P. aeruginosa*. The protease, hemolysis and elastase assays were performed in triplicates to assess the capability of the tested anti-diabetic drugs in sub-MIC to inhibit such enzymes. The one-way ANOVA test (Graphpad Prism 8 software) was employed to calculate the significant effects of sitagliptin, metformin and vildagliptin in sub-MIC on inhibition of these enzymes, *p* values < 0.05 were considered statistically significant. The data were presented as mean ± standard error of percentage change in protease, hemolysin or elastase production in anti-diabetic drug treated *P. aeruginosa* from untreated *P. aeruginosa* controls (Figure 2F–H, respectively). Sitagliptin and metformin significantly reduced the production of protease, hemolysin and elastase with percentages 48%, 85% and 56% and 43%, 76% and 26% in respective order. Taking into consideration that sitagliptin inhibited elastase production significantly when compared to metformin (*p* < 0.0001). Markedly, vildagliptin lacked the ability to inhibit the three enzymes.

#### 3.3.4. Inhibition of *P. aeruginosa* Pigment

Pyocyanin has emerged as an important *P. aeruginosa* virulence factor. Pyocyanin production was quantified in the absence and presence of anti-diabetic drugs in sub-MIC. The experiment was performed three times, and the one-way ANOVA test (Graphpad Prism 8 software) was used to compare between sitagliptin, metformin and vildagliptin in sub-MIC treated and untreated *P. aeruginosa*, and the results were considered statistically significant when *p* values < 0.05. The data were expressed as mean ± standard error of percentage change from untreated *P. aeruginosa* controls (Figure 2I). Although, vildagliptin did not decrease significantly the production of pyocyanin pigment, sitagliptin and metformin showed significant ability to reduce pyocyanin production. The inhibition percentages of pyocyanin pigment in presence of sitagliptin and metformin were 51% and 56%, respectively.

### 3.4. Resistance to Oxidative Stress

The resistance of *P. aeruginosa* to oxidative stress has a crucial role in its survival inside cells and invasiveness capability. The diameters of inhibition zones developed around discs loaded with hydrogen peroxide on LB agar plates containing anti-diabetic drugs in sub-MIC and streaked with *P. aeruginosa*, were measured. The one-way ANOVA test (Graphpad Prism 8 software) was used test the significance of sitagliptin, metformin and vildagliptin in sub-MIC effects on the resistance of *P. aeruginosa* to oxidative stress in comparison to untreated *P. aeruginosa*. The experiment was conducted in triplicate and results were considered statistically significant when *p* values < 0.05. The data obtained were presented as mean ± standard error of inhibition zones in mm of anti-diabetic treated and untreated *P. aeruginosa* controls (Figure 2J). The diameters of inhibition zones in plates with sitagliptin or metformin were significantly increased, which indicates their ability to reduce the tolerance of *P. aeruginosa* to oxidative stress. While, vildagliptin did not show a significant effect.

### 3.5. Down Regulation of P. aeruginosa QS Genes

QS genes’ expression levels were evaluated in control *P. aeruginosa* and in anti-diabetic (sub-MIC) treated *P. aeruginosa* by qRT-PCR using the 2^−ΔΔCt^ method (Figure 3). It is important to mention that the expression levels of the genes that encode the AIs of the three QS systems *lasI*, *rhlI* and *pqsA* and their receptors *lasR*, *rhlR* and *pqsR* in *P. aeruginosa* were significantly reduced when treated with sitagliptin, metformin or vildaglipyin in sub-MIC as compared to control untreated *P. aeruginosa*. The presented data are the mean ± standard error from three experiments, and *p* < 0.05 was considered significant using the one-way ANOVA test (Graphpad Prism 8 software. ns: non-significant, *p* > 0.05; *: *p* ≤ 0.05; **: *p* ≤ 0.01 and ***: *p* ≤ 0.001.). The expression levels of *lasI* and its receptor *lasR* were decreased three-and four-fold in the presence of sitagliptin or metformin, respectively, while vildgliptin decreased the expression of both genes about two-fold. Sitagliptin, metformin or vildgliptin in sub-MIC reduced the expression of the *rhlI* gene and its receptor *rhlR* gene three-to four-fold and two-to three-fold, respectively. Moreover, the expression levels of AI encoding gene *pqsA* and its receptor encoding gene *pqsR* were decreased three-to four-fold in presence of sitaliptin or metformin in sub-MIC and three-fold in presence of vildgliptin in sub-MIC.

### 3.6. In Vivo Protection Activity

The anti-diabetic drugs protection activity from *P. aeruginosa* pathogenesis was further assessed in vivo. Seven groups (each comprising five mice) were used, the survival of mice was reported every day over 5 successive days and plotted using the Kaplan–Meier method and significance (*p* < 0.05) was calculated using the Log-rank test, GraphPad Prism 8 (Figure 4). All mice in the two negative control groups survived, while only two survived out of five (40% survival) in the mice injected with DMSO-treated bacteria or untreated bacteria. Notably, all mice injected with *P. aeruginosa* treated with sitagliptin showed 100% survival, conferring 60% protection in comparison to mice injected with untreated *P. aeruginosa*. On the contrary, both metformin and vildgliptin failed in protecting mice from *P. aeruginosa* as only two of the five mice survived (60% death). These results obviously indicate that treatment of *P. aeruginosa* with sitagliptin in sub-MIC significantly decreased the bacterial capacity to kill mice (*p* = 0.0016) using the Log rank test for trend (GraphPad Prism 8), while metformin and vildgliptin did not reduce the capacity of *P. aeruginosa* to kill mice.

### 3.7. Docking of Anti-Diabetic Drugs to QS Receptors

To better understand how sitagliptin, metformin, and vildagliptin can inhibit *P. aeruginosa* QS, we performed molecular modeling and a docking study using the Schrodinger suite. In order to perform docking, the X-ray crystal structures of LasR, QscR, and pqsR were obtained from a protein data base as receptors (PDB: 1RO5, 6CC0, 6MVN). The docking study on LasR (PDB: 1RO5) is shown in Figure 5 and the rest are available in the Appendix A. Docking results explains the in vitro and in vivo results for the inhibitory activity, sitagliptin showed the highest inhibitory activity compared to metformin and vildagliptin, respectively, with the three receptors.

The modeling study revealed that the tetrazole ring of sitagliptin fit to the pocket forming hydrogen bonding with Arg-154. Moreover, the aromatic ring of sitagliptin was found to have pi–pi stacking with the sidechain Phe-117 and pi–cation with the Arg-30 sidechain (Figure 5A,B). The hydrophobic interactions were observed between the di-fluoro substitution on the aromatic ring of sitagliptin and the sidechain of Phe-105, Ala-106, and Ile-107. However, Metformin, ranked second in docking; it showed two hydrogen bonding between the amidine group and Arg-30 and Phe-105 sidechain of the receptor (Figure 5C,D). Lastly, vildagliptin which is characterize by amantadine moiety showed to be the least inhibitory drug. The modeling study suggested the amantadine group of vildagliptin is exposed to the solvent outside the active site, and a hydrogen interaction was formed between the keto- group and Arg-30 sidechain (Figure 5E,F). It was concluded that metformin and vildagliptin had fewer binding interactions whereas sitagliptin had multiple binding interactions and fits the active site which could explain its favorable inhibitory effect compared to the other drugs.

## 4. Discussion

*P. aeruginosa* is one of the most noteworthy human pathogens and has been recognized as a priority pathogen on the ESKAPE pathogen list [38]. It uses both competitive and cooperative strategies to conquer a variety of niches utilizing an expanded arsenal of virulence factors [3,14]. *P. aeruginosa* possesses almost all of the known antimicrobial resistance mechanisms which is why empirical antibiotic treatments are often ineffective [16]. As a consequence, finding novel alternative therapeutic approaches is a necessity, targeting the bacterial social behaviors involved in their pathogenesis is one of these promising approaches [3,39]. Targeting the bacterial cell-to-cell communication quorum sensing system (QS) has several merits, particularly it avoids direct any effect on bacterial growth, decreasing the emergence of bacterial resistance [3,20,39,40,41]. Our initial hypothesis was the repurposing of widely used drugs like those which are prescribed routinely for chronic diseases such as diabetes. Besides the advantages of the strategy of repurposing drugs [8], our previous work showed the effectiveness of sitagliptin against *Serratia marcescens* [6]. Bearing this in mind, we aimed in this work to test the anti-QS activities of some anti-diabetic drugs with diverse chemical structures against *P. aeruginosa* that may be helpful either in using these pharmaceuticals or even providing prospects for testing chemically related compounds.

Pyrazine derivatives efficiently down-regulated the QS of *Vibrio cholerae* via targeting the LuxO regulator, inhibited biofilm and diminished the bacterial adhesion and invasion onto the intestinal cell lines [42]. Triazole moiety-containing compounds are capable of binding several biological targets by hydrogen bonding and dipole interactions, and showed antibacterial activities such as those seen in tazobactam and cefatrizine [43,44]. Remarkably, triazole derivatives especially those comprising N-acyl L-homoserine lactone analogs, isoxazole and thymidine structures inhibited QS activities in *P. aeruginosa* [45] which is why they have been proposed as potential lead structures for the development of anti-QS agents [45,46]. In this direction, that pyrazine derivative with the triazole moiety anti-diabetic drug, sitagliptin, significantly attenuated the virulence of *Serratia marcescens* [6]. It was not the only anti-diabetic drug that has anti-QS activity, biguanide anti-diabetic drug metformin showed efficient anti-pseudomonal activity [11], and potentiated the antibacterial activities of gold nanoparticles [12]. Moreover, metformin augmented the bacterial biofilm eradication when formulated with gold nanoparticles [12] that may indicate its anti-QS activity. Pyrrolidine is another chemical moiety that was proven to have antifungal and antibacterial activity against *P. aeruginosa* and showed anti-QS activity [13]. Vildagliptin with its pyrrolidine chemical moiety may be a promising candidate to be tested as an anti-QS agent.

*P. aeruginosa* harbors three QS systems that regulate its virulence, two LuxI/LuxR type QS and one non-LuxI/LuxR type system called the *Pseudomonas* quinolone signal (PQS) system [3]. In the first QS system, the LuxI homolog LasI synthesizes the C12-homoserine lactone autoinducer that is detected by the cytoplasmic LuxR homolog LasR [47]. In the second Qs system, RhlI synthesizes another autoinducer, butanoyl homoserine lactone, that binds at high concentrations to RhlR, a second LuxR homolog [48]. Furthermore, QscR is an orphan homolog for LuxR that does not have a partner LuxI homolog but binds to the autoinducers produced by LasI [49]. PQS is an additional non-LuxI/LuxR QS system that is produced by PqsA, B, C, D, and H and is detected by the regulator PqsR (also named MvfR) [50]. In this study, it was shown that three tested anti-diabetic drugs (in sub-MIC) down regulated the expression of both autoinducers of the three QS systems encoding genes *lasI*, *rhlI* and *pqsA* and also their receptors’ encoding genes *lasR*, *rhlR* and *pqsR* in *P. aeruginosa*.

After testing the capability of tested drug to down regulate the QS encoding genes, it was necessary to study their effect on the phenotypic behavior of *P. aeruginosa*. In this work, we evaluated the effect of tested drugs on *P. aeruginosa* in sub-MIC (0.1 MIC) to avoid their effect on bacterial growth. Indeed, *P. aeruginosa* harbors an arsenal of virulence factors that are regulated and organized by QS in different stages of infection [3]. The Violacein stain is produced as secondary metabolite associated with biofilm production and regulated by QS [25]. Because of the easy visualization of violacein, it has become a useful indicator of QS inducers and their inhibitors [51]. Significantly, sitagliptin and metformin reduced the production of violacein, while vildagliptin failed. Biofilm formation and bacterial motilities are regulated by QS and play a crucial role in *P. aeruginosa* pathogenesis [15,26,28]. Sitagliptin and metformin significantly inhibited the biofilm formation and swimming, swarming and twitching motilities, while vildagliptin did not show significant inhibition. *P. aeruginosa* produces a wide range of enzymes which tremendously enhance the bacterial invasiveness [3,15]. In this study, sitagliptin and metformin reduced significantly the production of protease, elastase and hemolysin, meanwhile vildagliptin did not show any significant effect. Pyocyanin is a *P. aeruginosa* characterized pigment, and it has the ability to oxidize and reduce other molecules, killing competing microbes as well as mammalian cells [52]. Although vildagliptin did not reduce pyocyanin production, sitagliptin and metformin significantly succeeded in the reduction of pyocyanin production. QS orchestrates *P. aeruginosa* virulence, allowing it to counteract the immune response by increasing its tolerance to the generated oxidative stress inside immune cells’ phagosomes [53]. Sitagliptin and metformin significantly reduced the tolerance of *P. aeruginosa* to oxidative stress, while vildgliptin did not show any effect. Noteworthily, sitagliptin significantly inhibited bacterial motilities and elastase enzyme production when compared to metformin.

The three drugs were tested to investigate their protective effect against *P. aeruginosa* pathogenesis by the mice survival in vivo model [36]. Interestingly, 60% protection was conferred to mice injected with sitagliptin (sub-MIC) treated *P. aeruginosa*, showing a trend for better survival in comparison to mice injected with untreated bacteria or DMSO treated bacteria. In accordance with in vitro findings, the in vivo results declared that sitagliptin can serve as an efficient *P. aeruginosa* QS-inhibitor. These results are consistent with our previous data in which sitagliptin efficiently reduced the virulence of *Serratia marcescens* [6]. In agreement with the in vitro results, vildagliptin did not show protection for mice from *P. aeruginosa*. However, the outcomes of in vitro and in vivo studies do not contradict each other in case of sitagliptin and vildagliptin, metformin “in contrast to in vitro studies” failed to protect mice.

For further understanding, the molecular docking study was performed to confirm the virtual ability of the tested drugs to bind to the three LuxR homologs LasR, QscR and PqsR “QS receptors”. The sitagliptin high docking scores indicate a promising antagonistic activity that means an efficient interference with the binding of the autoinducers to their receptors, in accordance with both in vitro and in vivo findings. Sitagliptin fully occupies all the necessary required receptors’ active sites, particularly in the binding of its aromatic ring to phenyl alanine-117 site. On the other hand, the planner nature of the bulky aliphatic adamantyl group of vildagliptin with its rapid conformational changes was stuck as a barrier while fitting on QS receptors. As a consequence, the least docking scores to QS receptors were observed with vildagliptin which may explain its ineffectiveness in spite of its capability to down-regulate QS encoding genes. In between, the small very active biguanide moiety of metformin, does not enable it to fit all the active pockets on the QS receptors. The metformin docking scores into QS receptors indicated promising antagonistic activity but lower than that showed by sitagliptin. These findings are in compliance with in vitro results which showed efficient activity of metformin as anti-QS, but also it was less effective than sitagliptin especially in inhibition of bacterial motilities and elastase enzyme. In case of metformin, in vitro studies are promising, but subsequent in vivo studies fail to show any efficacy which may be owed to its chemical nature. Meanwhile the sitagliptin is stable in a wider range of pH which explains its stability even in acidic urine [54]; metformin activity is owed to its unionized form and is very sensitive to acidic pH and rendered ionized in its inactive form. Notably, the *Pseudomonas* growth decreases the pH of the surrounding media during its growth [55] which does not allow metformin in active form to inhibit QS sufficiently to diminish *P. aeruginosa* pathogenesis in vivo. Based on the above, we can suggest that the anti-QS activity of sitagliptin, is owed to its capability to occupy and block the QS receptors compared to metformin and vildagliptin. Bearing in mind that the three drugs downregulated the QS encoding genes, we suppose the receptor inactivation/signaling blockage may play the most important role in diminishing the QS activities.

## 5. Conclusions

Finally, it can be concluded that targeting QS with its crucial roles in the controlling of virulence is a promising approach to diminish bacterial pathogenesis. The merits of this approach are that it does not induce the emergence of resistance because no pressure is exerted on bacterial growth and attenuates bacteria which enhances the immunity to their eradication. The merits will be magnified when using already approved safe drugs as anti-diabetics. In this study we provide an insight into the anti-QS activities of sitagliptin, metformin and vildagliptin on *P. aeruginosa*. We showed that sitagliptin could serve as efficient, target specific and safe anti-QS in vitro and in vivo. Moreover, we propose a pharmacophore for anti-QS future drugs.

## Figures and Tables

**Figure 1 microorganisms-08-01285-f001:**
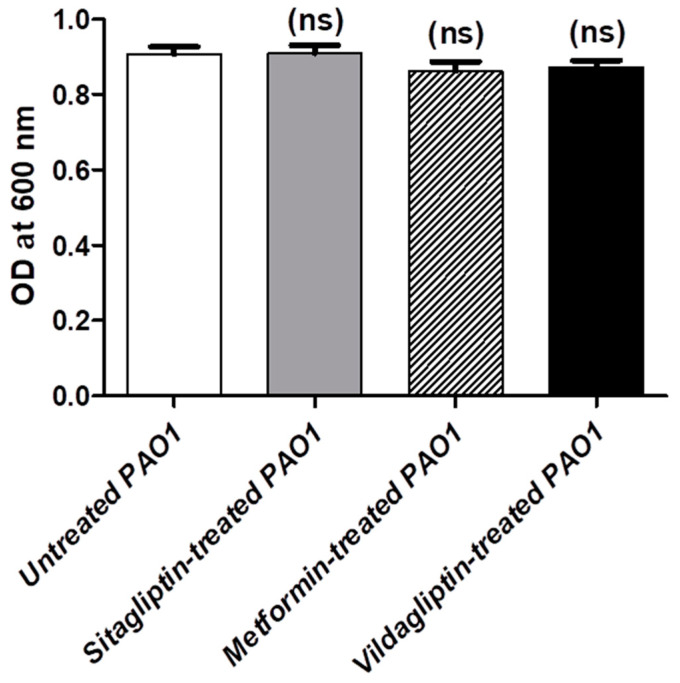
Effect of anti-diabetic drugs in sub-minimum inhibitory concentration (MIC) on growth of *P. aeruginosa*. The optical densities of *P. aeruginosa* cultures treated or untreated with anti-diabetic drugs in sub-MIC were measured at 600 nm. A one-way ANOVA test was used to compare between sitagliptin, metformin and vildagliptin treated and untreated cultures and the results were considered statistically significant when *p* values < 0.05. There was no significant difference between an OD600 of all the anti-diabetic drugs treated and untreated cultures after overnight incubation in Luria–Bertani (LB) broth (*p* = 0.3286). ns: nonsignificant, *p* > 0.05.

**Figure 2 microorganisms-08-01285-f002:**
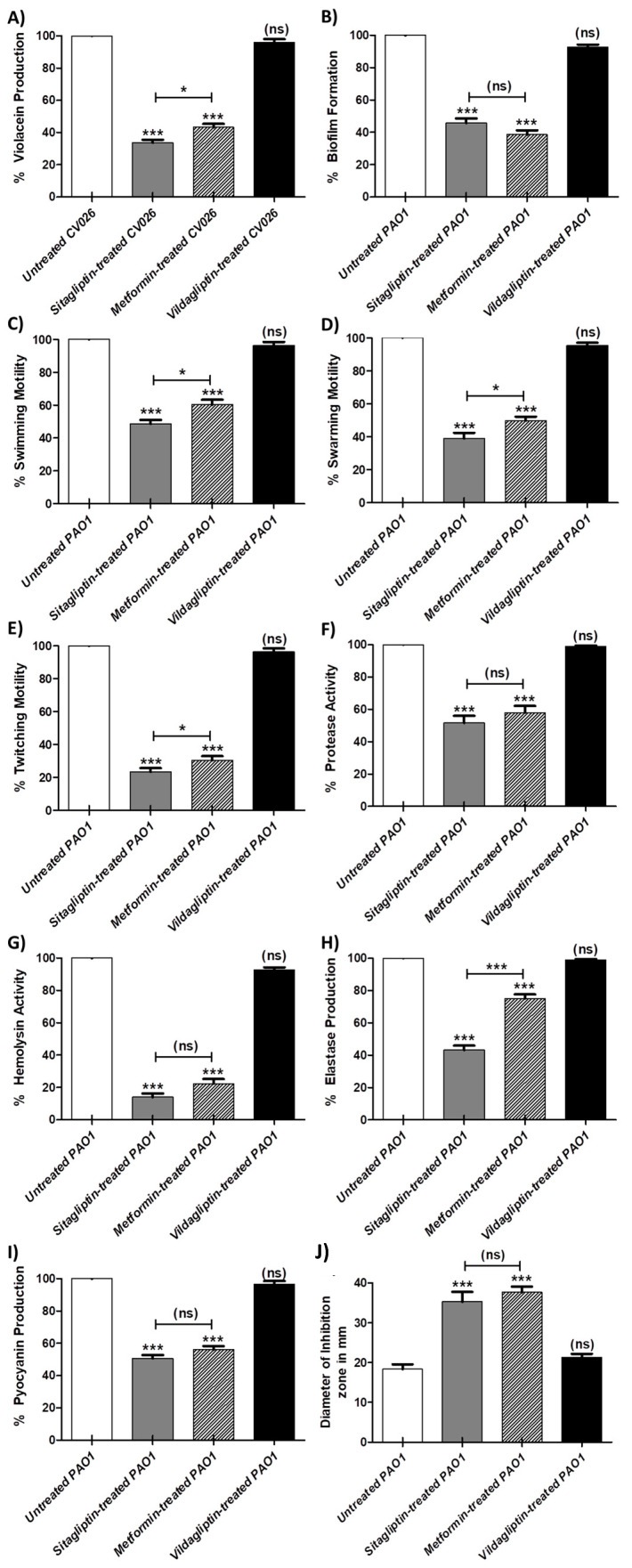
Inhibitory effect of the tested anti-diabetic drugs on bacterial virulence. Bacterial cultures were prepared with and without 1/10 MIC of sitagliptin, metformin or vildagliptin. The one-way ANOVA test (Graphpad Prism 8 software) was used to compare between the obtained results of sitagliptin, metformin and vildgliptin in sub-MIC treated and untreated bacterial cultures. The results were considered statistically significant when *p* values < 0.05. The data obtained were presented as mean ± standard error of percentage change from untreated cultures. (**A**) Violacein Production: Significantly, sitagliptin and metformin inhibited violacein production (*p* < 0.0001), while vildgliptin did not show significant inhibition (*p* = 0.1272). (**B**) Biofilm formation: Sitagliptin and metformin significantly interfered with the biofilm formation (*p* < 0.0001), in contrast vildgliptin did not interfere with biofilm formation (*p* = 0.0686). (**C**–**E**) Motilities: Vildagliptin did not show significant reduction in *P. aeruginosa* swimming, swarming or twitching motility (*p* = 0.1702, 0.0573 and 0.1689, respectively). Meanwhile, sitagliptin and metformin decreased the *P. aeruginosa* motilities (*p* < 0.0001). (**F**–**H**) Enzyme production: Both sitagliptin and metformin reduced significantly the production of protease, hemolysin and elastase, respectively (*p* < 0.0001), while vildagliptin did not show a significant effect (*p* = 0.1599, 0.0911 and 0.1289, respectively). (**I**) Pyocyanin production: Vildgaliptin did not reduce the pigment production (*p* = 0.1471), while sitagliptin and metformin significantly reduced the production of pyocyanin (*p* < 0.0001). (**J**) Tolerance to oxidative stress: the presented data are mean ± standard error of inhibition zones in mm of anti-diabetic treated and untreated *P. aeruginosa* control. Sitagliptin and metformin significantly reduced the resistance of *P. aeruginosa* to hydrogen peroxide (*p* < 0.0001). On the other hand, vildagliptin did not show any significant activity (*p* = 0.1145). ns: non-significant, *p* > 0.05; *: *p* ≤ 0.05 and ***: *p* ≤ 0.001.

**Figure 3 microorganisms-08-01285-f003:**
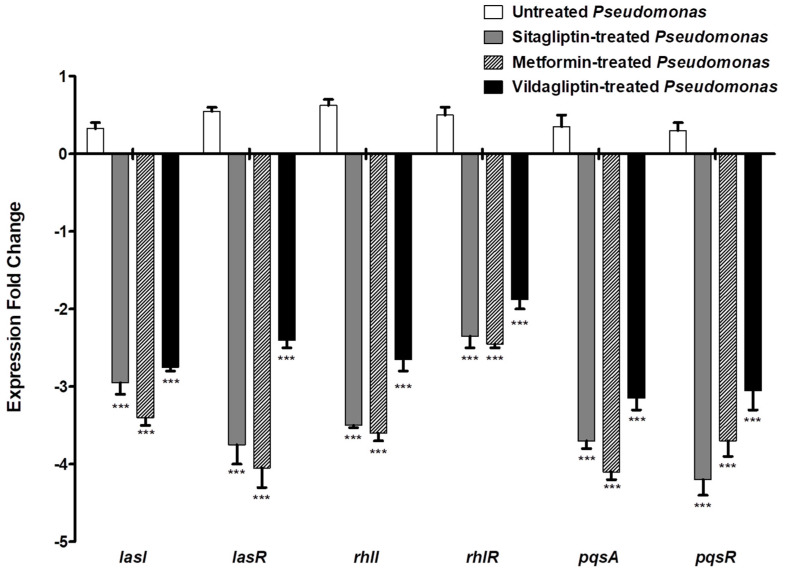
Down-regulation of *P. aeruginosa* QS genes. RNA was isolated from *P. aeruginosa* cultures treated and untreated with anti-diabetic drugs in sub-MIC to be used in cDNA synthesis. The cDNA was amplified by qRT-PCR and changes in the expression of each QS gene that were normalized in relation to the mean critical threshold values of housekeeping gene *rpoD*. Expression fold change in gene expression in anti-diabetic treated *P. aeruginosa* was calculated by the 2^−ΔΔCT^ method and compared to untreated bacteria. The data shown are the mean ± standard errors from three experiments. *p* < 0.05 was considered significant using the one-way ANOVA test. Three tested drugs significantly decreased the expression of genes *lasI*, *lasR*, *rhlI*, *rhlR*, *pqsA* and *pqsR* (*p* < 0.0001). ***: *p* ≤ 0.001.

**Figure 4 microorganisms-08-01285-f004:**
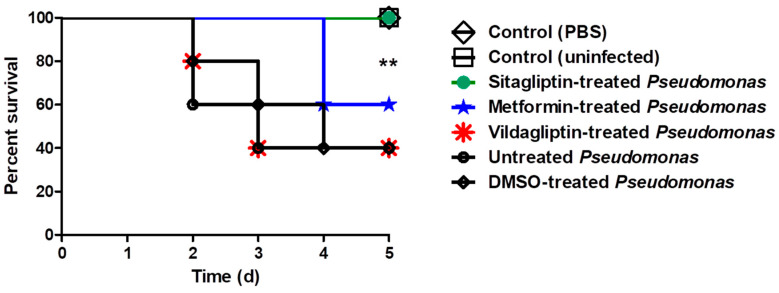
In-vivo protection from *P. aeruginosa***.** Seven groups composed of 5 healthy female mice were used. Group 1, mice were intraperitoneally injected with sitagliptin (sub-MIC) treated *P. aeruginosa* in sterile phosphate buffer saline (PBS), group 2 were injected with metformin (sub-MIC) treated *P. aeruginosa,* group 3 were injected with vildagliptin (sub-MIC) treated *P. aeruginosa,* group 4 was injected with dimethyl sulfoxide (DMSO)-treated *P. aeruginosa*, group 5 were injected with untreated *P. aeruginosa*, group 6 were injected with sterile PBS and group 7 were left un-inoculated. Mice survival in each group was recorded every day over 5-days, plotted using the Kaplan–Meier method and significance (*p* < 0.05) was calculated using the Log rank test, GraphPad Prism 8. All mice in negative control groups (groups 6 and 7) survived, while only 40% of mice survived (2 out of 5 mice) in the groups comprising metformin treated bacteria, vildagliptin treated bacteria, DMSO-treated bacteria or untreated bacteria. On the other hand, all mice injected with sitagliptin-treated *P. aeruginosa* survived, showing 100% survival, conferring 60% protection (Log rank test for trend *p* = 0.0016). **: *p* ≤ 0.01.

**Figure 5 microorganisms-08-01285-f005:**
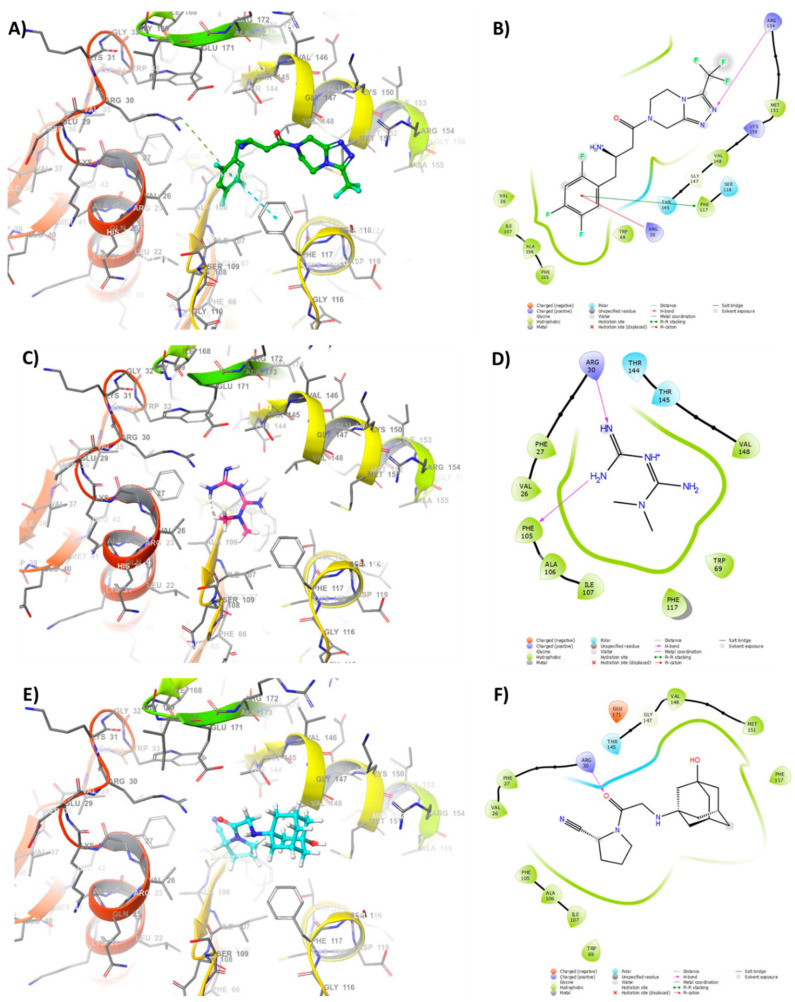
Docking study of tested anti-diabetic drugs into *P. aeruginosa* QS receptor LasR. 3D binding mode of (**A**) sitagliptin (green), (**C**) metformin (purple) and (**E**) vildagliptin (cyan) with LasR. 2D representation of (**B**) sitagliptin, (**D**) metformin and (**F**) vildagliptin binding interactions with amino acids in the active sites of LasR.

**Table 1 microorganisms-08-01285-t001:** Sequences of the used primers in this study.

Target Gene	Sequence (5′–3′)	Reference
*lasI*	For: CTACAGCCTGCAGAACGACA	[35]
	Rev: ATCTGGGTCTTGGCATTGAG	
*lasR*	For: ACGCTCAAGTGGAAAATTGG	[35]
	Rev: GTAGATGGACGGTTCCCAGA	
*rhlI*	For: CTCTCTGAATCGCTGGAAGG	[35]
	Rev: GACGTCCTTGAGCAGGTAGG	
*rhlR*	For: AGGAATGACGGAGGCTTTTT	[35]
	Rev: CCCGTAGTTCTGCATCTGGT	
*pqsA*	For: TTCTGTTCCGCCTCGATTTC	[34]
	Rev: AGTCGTTCAACGCCAGCAC	
*pqsR*	For: AACCTGGAAATCGACCTGTG	[35]
	Rev: TGAAATCGTCGAGCAGTACG	
*rpoD*	For: GGGCGAAGAAGGAAATGGTC	[34]
	Rev: CAGGTGGCGTAGGTGGAGAAC	

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
