# Peer review of "Repurposing Anti-diabetic Drugs to Cripple Quorum Sensing in Pseudomonas aeruginosa"

_microorganisms, 2020, doi:10.3390/microorganisms8091285_

Round 1
Reviewer 1 Report
The manuscript fits the scope of the journal.
I recommend having the text checked by a native English speaker. In too many instances, articles are missing (a, the) and/or are in excess. In addition, conjunctions are sometimes used incorrectly. Furthermore, the punctuation sometimes is incorrect (usage of semicolon instead of colon for instance). Please indicate centrifugation speeds in g instead of rpm. Please respect space character between value and unit (e.g. 25 nm; 30 min).
Abstract
Line 24, and it possess
Line 28, repurpose the
Line 32, Although the three investigated drugs
Line 34, replace “however” by “in contrast”
Introduction
Line 49, clinically-relevant?
Line 65, strategy is that targeting
Line 75, which is widely
Line 76, as the most common
Line 79, that a pyrazine
Line 83, repurposing the
Materials and methods
Line 87, the model
Line 88, in the
Line 99, density of approximately
Line 103, and many more times in the text, e.g. Lines 110, 125: 0.x AU
Line 114, by adding 100
Lines 128-130, several space characters are missing
Line 143, has been performed in triplicate
Line 145, standard error
Lines 153, 164, 180, and more, space characters are missing
Line 217, which wavelength?
Line 223, using the
Line 251, mice were held/handled?
Results
Line 263, vildagliptin with x, y, and z mg/mL, respectively.
Line 267, One-way ANOVA (…) was employed
Line 269, results were considered significant if P < 0.05
Line 282, the QS activity
Line 287, results were considered significant if P < 0.05
Line 306, In contrast, …
Figure captions and text, sometimes there is too much redundant information (in the captions)
Line 328, In contrast, …
Line 410 as only 2 of the 5 mice survived
Line 416, a docking study
Line 420, sitagliptin showed the highest inhibitory activity compared to metformin and vildagliptin …
Discussion
Line 468, Pds efficiently down-regulated the
Line 530, that means an efficient
Line 537, in between?
Line 540, are in compliance with
Lines 543-544, On the other hand, sitagliptin is stable in a wider range of
Some very recent studies on anti-quorum sensing activity of certain substances against Pseudomonas aeruginosa should be cited (e.g. Rafiee F. et al., 2020, AMB Express 10, 82; Gholami M., et al., 2020, AMB Express 10, 111; Ouyang, J., 2020, Microb Pathog 149, 104291).
Author Response
Dear Reviewer,
We greatly appreciate your efforts and will did all the points you raised.
Please see the attachment.
Best Regards,
Authors

Reviewer 2 Report
Authors of the paper entitled “Repurposing anti-diabetic drugs to cripple quorum sensing in Pseudomonas aeruginosa”, by Hegazy et al., aimed to use anti-diabetic drugs sitagliptin, metformin and vildagliptin as anti-QS in P. aeruginosa. The effects of their sub-inhibitory concentrations on the expression of QS-encoding genes and QS-regulated P.a. virulence factors were assessed. The topic is attractive, up-to-date and important, but I'm not sure what's new in this study? How far-reaching new elements appear in this work compared to the previous publication (No. 6). The question remains whether and what cognitive and practical significance the authors' observations may have.
Introduction section presents and justifies the purpose of research in very general terms. There is no so-called "fluidity" of the text. In addition to examining the impact of drugs selected without substantive justification on selected characteristics of the selected species of bacteria, there is no so-called leading thought/significant keynote of the study.
It is possible to propose a change in the organization of the text - moving the fragment L 68-82 before L 54-67. This may provide the opportunity for a clearer rationale for the selection for testing of anti-diabetic drugs and these bacteria, which are a common cause of infections in immunocompromised individuals (e.g., diabetics). May be the authors have another justification?
Currently the justification for the study is = research was undertaken because earlier the authors were able to demonstrate (in an identical in vitro experiments) the anti-QS effectiveness of sitagliptin against Serratia marcescens (ref. No 6, Abbas and Hegazy, 2020).
Material and methods section
A broad set of experiments, as listed in subsections 2.1. - 2.11., was basically correctly described.
However, several questions, doubts and criticisms are listed below:
- L 94-95 There is no description of the preparation of drugs for series of individual experiments (CAS DataBase Reference, solvents, storage conditions for solution, time of storage and stability, etc.). Especially since it concerns the key preliminary experiments determining the concentration of drugs used during the entire research cycle.
- L 94-99 - MICs of the drugs were tested using agar dilution method and/or microbroth dilution method. It is well known that the bioavailability of an antimicrobial agent is closely dependent on the physicochemical form of the drug and the microenvironmental test conditions. In this work, some experiments on the phenotypic features of P. aeruginosa were carried out in the liquid environment - broth (2.3; 2.4; 2.7-2.9.). The rest - in the agar dilution.
Do the authors have certainty that the MICs of the investigated drugs are identical in both types of experimental conditions (?). It is necessary to present not only intermediate data (Fig. 1, Fig. 2), but e.g. photographic documentation of the control conditions of the experiments described in sections 2.5; 2.6.; 2.10. Do the authors could comment on it.
- what the final percentage of DMSO (drug solvent) concentration was used as a control at each stage of the study (I suggest sharing "raw" data).
- L 100-106 - question is why 1/10 of the MICs determined in an agar-dilution method were used as standard concentrations in further experiments (in agar-condition and broth-condition). Data or even comments on it are necessary.
- L 122-125 - how technically is possible to..."Aliquots of 10 μl of the P. aeruginosa suspension were adjusted to OD600 of 0.4...."
- L 121; 133; 148; 159; 174; 187; 199 - These equations should be in the required format, available in Word processor.
- L 150-152 (13,000rpm for 20 min); 161-163 (13,000rpm for 20min) ; 179-180 (13,000rpm for 20min); 211-212 (7000 rpm for 15min); - Uniform or different treatment conditions for the bacteria under study drugs should necessarily be given. Only in this way can it be ensured that the experimental conditions set out are reproduced reliably.
I suggest that in order to avoid repetition of descriptions in individual subsections, a universal diagram / table / figure etc. could be prepared.
However, a much more serious critical remark is, according to my opinion, an incorrectly formulated plan for a serie of in vivo experiments (subsection 2.12), which can be considered as the most important subsection. A note on this matter will be given below.
Questions, for example:
- According to my experience - it is not mice infected with "wild" P. aeruginosa or with these bacteria treated with anti-diabetic drugs in vitro, that should be analyzed.
Such an experiment plan is in no way appropriate to reality.
To adequately reflect the real in vivo conditions (infection and experimental potential therapy), a systemic or localized infection (P. aeruginosa) should be induced in the experimental animals (e.g. mice) and these selected anti-diabetic drugs (at low concentration - 1/10 MIC as the authors decided) should be used for "therapy". Only this may give an answer (yes or no) whether the expected results of limiting the QS-regulated expression of genes encoding important virulence factors (what and to what extent? - this remains to be determined) will translate into therapeutic effects in vivo!!!.
The currently presented research results in the above field are of a very preliminary nature.
I must admit that the authors in this work faced, given the scope and manner of execution, an interesting task, but not yet feasible. Or they will treat the obtained results as the result of promising preliminary studies drawing further research plans. In the latter case - they will change the title of the work, purpose and final conclusions.
Results section
Presentation of the results is generally correct, of course according to the performed experiments. However ----
- For experiments described in Material & Methods in L 134-159 - graphical documentation would be expected.
- The organization and the legend of Fig. 2., need to be changed. Perhaps it should be divided into several smaller ones and the proper legend (short description of the experiments, previously given in M&M) should be separated from the commentary on the obtained results.
Currently, I do not undertake to submit other substantive comments, because I see numerous critical remarks concerning experimental protocols, main idea etc. This part should be rewritten after correcting the description of the Materials and Methods section and or provide an additional data.
Discussion and Conclusions sections
L 549-550 - I agree with the authors' statement that ..."Finally, it can be concluded that targeting QS with its crucial roles in controlling of virulence is a promising approach to diminish bacterial pathogenesis". This is in line with the knowledge accepted over 20 years ago. Most statements, however, are very simplified and overinterpreted the obtained test results.
L 550-556: phrase:..."The merits will be magnified when using already approved safe drugs as anti-diabetics. In this study we provide an insight on the anti-QS activities of sitagliptin, metformin and vildagliptin on P. aeruginosa. We showed that sitagliptin could serve as efficient, target specific and safe anti-QS in-vitro and in-vivo. Moreover, we propose a pharmacophore for anti-QS future drugs.
Concerning the above - some questions are arised:
- how the authors imagine using these drugs for the intended purpose. in real conditions of local or systemic infection caused by Pseudomonas aeruginosa (?), in people with diabetes treated e.g. with metformin or sitagliptin or in non-diabetics with the above-mentioned infection types, glycemic control.
- what is the relation of the concentrations of drugs proposed by the authors that are effective as limiting the functioning of QS versus therapeutic in diabetes?
- what QS disruption strategy we are dealing with in this case: only receptor inactivation, either signals synthesis inhibition, or signals degradation/signaling blockage. It can be suggested to deepen the interesting considerations presented in the fragment of the Discussion (L 528-548). On the other hand, for the sake of clarity of the Discussion text, I suggest moving some general information to the Introduction.
These and other mentioned issues should be discussed in depth.
Other remarks
- It is difficult to accept the use of an editorial character like (;) in creating sentences throughout the text. I suggest you omit them and replace with an appropriate one.
- The format of the list of references seems to need improvement.
- L 468 - Vibrio cholerae not Vibrio cholera
Author Response
Dear Reviewer 2,
We are very thankful for you and appreciate you efforts which give us more ideas and improvements in this work and in future work. We were happy to answer the raised points and tried to clarify each single point.
Please see the attachment.
Best Regards,
Authors

Round 2
Reviewer 2 Report
Although not all of my remarks and comments were taken into account, I accept the amendments and additions to the text. I have no more serious comments.